# Representation Learning for Treatment Effect Estimation from Observational Data

**Liuyi Yao**
SUNY at Buffalo
liuyiyao@buffalo.edu

**Sheng Li**
University of Georgia
sheng.li@uga.edu

**Yaliang Li**
Tencent Medical AI Lab
yaliangli@tencent.com

**Mengdi Huai**
SUNY at Buffalo
mengdihu@buffalo.edu

**Jing Gao**
SUNY at Buffalo
jing@buffalo.edu

**Aidong Zhang**
SUNY at Buffalo
azhang@buffalo.edu

## Abstract

Estimating individual treatment effect (ITE) is a challenging problem in causal inference, due to the missing counterfactuals and the selection bias. Existing ITE estimation methods mainly focus on balancing the distributions of control and treated groups, but ignore the local similarity information that provides meaningful constraints on the ITE estimation. In this paper, we propose a local similarity preserved individual treatment effect (SITE) estimation method based on deep representation learning. SITE preserves local similarity and balances data distributions simultaneously, by focusing on several hard samples in each mini-batch. Experimental results on synthetic and three real-world datasets demonstrate the advantages of the proposed SITE method, compared with the state-of-the-art ITE estimation methods.

## 1 Introduction

Estimating the causal effect of an intervention/treatment at individual-level is an important problem that can benefit many domains including health care [12, 1], digital marketing [6, 34, 24], and machine learning [10, 37, 21, 23, 20]. For example, in the medical area, many pharmaceuticals companies have developed various anti-hypertensive medicines and they all claim to be effective for high blood pressure. However, for a specific patient, which one is more effective? Treatment effect estimation methods are necessary to answer the above question, and it leads to better decision making. Treatment effect could be estimated at either the group-level or individual-level. In this paper, we focus on the individual treatment effect (ITE) estimation.

Two types of studies are usually conducted for estimating the treatment effect, including the *randomized controlled trials (RCTs)* and *observational study*. In *RCTs*, the treatment assignment is controlled, and thus the distributions of treatment and control groups are known, which is a desired property for treatment effect estimation. However, conducting RCTs is expensive and time-consuming, sometimes it even faces some ethical issues. Unlike RCTs, *observational study* directly estimates treatment effects from the observed data, without any control on the treatment assignment. Owing to the easy access of observed data, observational studies, such as the potential outcome framework [27] and causal graphical models [26, 35], have been widely applied in various domains [15, 38, 12].

Estimating individual treatment effect from observational data faces two major challenges, *missing counterfactuals* and *treatment selection bias*. ITE is defined as the expected difference between the treated outcome and control outcome. However, a unit can only belong to one group, and thus the outcome of the other treatment (i.e., counterfactual) is always missing. Estimating counterfactual

outcomes from observed data is a reasonable way to address this issue. However, selection bias makes it more difficult to infer the counterfactuals in practice. For instance, in the uncontrolled cases, people have different preferences to the treatment, and thus there could be considerable distribution discrepancy across different groups. The distribution discrepancy further leads to an inaccurate estimation of counterfactuals.

To overcome the above challenges, some traditional ITE estimation methods use the treatment assignment as a feature, and train regression models to estimate the counterfactual outcomes [11]. Several nearest neighbor based methods are also adopted to find the nearby training samples, such as k-NN [8], propensity score matching [27], and nearest neighbor matching through HSIC criteria [5]. Besides, some tree and forest based methods [7, 33, 4, 3] view the tree and forests as an adaptive neighborhood metric, and estimate the treatment effect at the leaf node. Recently, representation learning approaches have been proposed for counterfactual inference, which try to minimize the distribution difference between treated and control groups in the embedding space [30, 18].

State-of-the-art ITE estimation methods aim to balance the distributions in a global view, however, they ignore the local similarity information. As similar units shall have similar outcomes, it is of great importance to preserve the local similarity information among units during representation learning, which decreases the generalization error in counterfactual estimation. This point has also been confirmed by nearest neighbor based methods. Unfortunately, in recent representation learning based approaches, the local similarity information may not be preserved during distribution balancing. On the other hand, nearest neighbor based methods only consider the local similarity, but cannot balance the distribution globally. Our proposed method combines the advantages of both of them.

In this paper, we propose a novel local **s**imilarity preserved **i**ndividual **t**reatment **e**ffect estimation method (**SITE**) based on deep representation learning. SITE maps mini-batches of units from the covariate space to a latent space using a representation network. In the latent space, SITE preserves the local similarity information using the Position-Dependent Deep Metric (PDDM), and balances the data distributions with a Middle-point Distance Minimization (MPDM) strategy. PDDM and MPDM can be viewed as a regularization, which helps learn a better representation and decrease the generalization error in estimating the potential outcomes. Implementing PDDM and MPDM only involves triplet pairs and quartic pairs of units respectively from each mini-batch, which makes SITE efficient for large-scale data. The proposed method is validated on both synthetic and real-world datasets, and the experimental results demonstrate its advantages brought by preserving the local similarity information.

## 2 Methodology

### 2.1 Preliminary

Individual treatment effect (ITE) estimation aims to examine whether a treatment $T$ affects the outcome $Y^{(i)}$ of a specific unit $i$. Let $\mathbf{x}_i \in \mathcal{R}^d$ denote the pre-treatment covariates of unit $i$, where $d$ is the number of covariates. $T_i$ denotes the treatment on unit $i$. In the binary treatment case, unit $i$ will be assigned to the control group if $T_i = 0$, or to the treated group if $T_i = 1$.

We follow the potential outcome framework proposed by Neyman and Rubin [29, 31]. If the treatment $T_i$ has not been applied to unit $i$ (also known as the out-of-sample case [30]), $Y_0^{(i)}$ is called the potential outcome of treatment $T_i = 0$ and $Y_1^{(i)}$ the potential outcome of treatment $T_i = 1$. On the other hand, if the unit $i$ has already received a treatment $T_i$ (i.e., the within-sample case [30]), $Y_{T_i}$ is the factual outcome, and $Y_{1-T_i}$ is the counterfactual outcome. In observational study, only the factual outcomes are available, while the counterfactual outcomes can never been observed.

The individual treatment effect on unit $i$ is defined as the difference between the potential treated and control outcomes[1]:

$$\mathbf{ITE}_i = Y_1^{(i)} - Y_0^{(i)}. \tag{1}$$

The challenge to estimate $\mathbf{ITE}_i$ lies on how to estimate the missing counterfactual outcome. Existing counterfactual estimation methods usually make the following important assumptions [17].

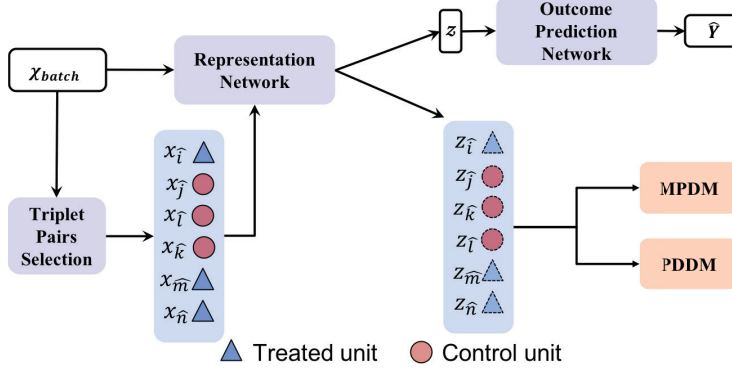

Figure 1: Framework of similarity preserved individual treatment effect estimation (SITE).

**Assumption 2.1** *(SUTVA). The potential outcomes for any unit do not vary with the treatment assigned to other units, and, for each unit, there are no different forms or versions of each treatment level, which lead to different potential outcomes [17].*

**Assumption 2.2** *(Consistency). The potential outcome of treatment $t$ equals to the observed outcome if the actual treatment received is $t$.*

**Assumption 2.3** *(Ignorability). Given pretreatment covariates $\mathbf{X}$, the outcome variables $Y_0$ and $Y_1$ is independent of treatment assignment, i.e., $(Y_0, Y_1) \perp\!\!\!\perp T|X$.*

Ignorability assumption makes the ITE estimation identifiable. Though it's hard to prove the satisfaction of the assumption, the researchers can make the assumption more plausible if the pretreatment covariates include the variables that affect both the treatment assignment and the outcome as much as possible. This assumption is also called "*no unmeasured confounder*".

**Assumption 2.4** *(Positivity). For any set of covariates $\mathbf{x}$, the probability to receive treatment $0$ or $1$ is positive, i.e., $0 < P(T = t|X = \mathbf{x}) < 1, \forall t$ and $\mathbf{x}$.*

This assumption is also named as population overlapping [9]. If for some values of $X$, the treatment assignment is deterministic (i.e., $P(T = t|X = \mathbf{x}) = 0$ or $1$), we would lack the observations of one treatment group, such that the counterfactual outcome is unlikely to be estimated. Therefore, positivity assumption guarantees that the ITE can be estimated.

## 2.2 Motivation

Balancing distributions of control group and treated group has been recognized as an effective strategy for counterfactual estimation. Recent works have applied distribution balancing constraints to either the covariate space [16] or latent space [18, 23].

Moreover, we assume that similar units would have similar outcomes. This assumption has been well justified in many classical counterfactual estimation methods such as the nearest neighbor matching. To satisfy this assumption in the representation learning setting, the local similarity information should be well preserved after mapping units from the covariate space $\mathcal{X}$ to the latent space $\mathcal{Z}$. One straightforward solution is to add a constraint on similarity matrices constructed in $\mathcal{X}$ and $\mathcal{Z}$. However, constructing similarity matrices and enforcing such a "global" constraint is very time and space consuming, especially for a large amount of units in practice. Motivated by the hard sample mining approach in the image classification area [14], we design an efficient local similarity preserving strategy based on triplet pairs.

## 2.3 Proposed Method

We propose a local **s**imilarity preserved **i**ndividual **t**reatment **e**ffect estimation (**SITE**) method based on deep representation learning. The key idea of SITE is to map the original pre-treatment covariate space $\mathcal{X}$ into a latent space $\mathcal{Z}$ learned by deep neural networks. Particularly, SITE attempts to enforce two special properties on the latent space $\mathcal{Z}$, including the *balanced distribution* and *preserved similarity*.

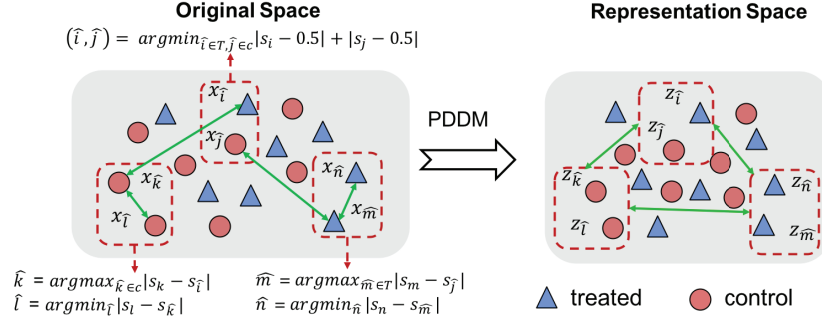

Figure 2: Triple pairs selection for PDDM in a mini-batch.

The framework of SITE is shown in Figure 1, which contains five major components: representation network, triplet pairs selection, position-dependent deep metric (PDDM), middle point distance minimization (MPDM), and the outcome prediction network. To improve the model efficiency, SITE takes input units in a mini-batch fashion, and triplet pairs could be selected from every mini-batch. The representation network learns latent embeddings for the input units. With the selected triplet pairs, PDDM and MPDM are able to preserve the local similarity information and meanwhile achieve the balanced distributions in the latent space. Finally, the embeddings of mini-batch are fed forward to a dichotomous outcome prediction network to get the potential outcomes.

The loss function of SITE is as follows:

$$\mathcal{L} = \mathcal{L}_{\text{FL}} + \beta \mathcal{L}_{\text{PDDM}} + \gamma \mathcal{L}_{\text{MPDM}} + \lambda ||W||_2, \tag{2}$$

where $\mathcal{L}_{\text{FL}}$ is the factual loss between the estimated and observed factual outcomes. $\mathcal{L}_{\text{PDDM}}$ and $\mathcal{L}_{\text{MPDM}}$ are the loss functions for PDDM and MPDM, respectively. The last term is $L_2$ regularization on model parameters $W$ (except the bias term).

Next, we describe each component of SITE in detail.

### 2.3.1 Representation Network

Inspired by [18], a standard feed-forward network with $d_h$ hidden layers and the rectified linear unit (ReLU) activation function is built to learn latent representations from the pre-treatment covariates. For the unit $i$, we have $\mathbf{z}_i = f(\mathbf{x}_i)$, where $f(\cdot)$ denotes the representation function learned by the deep network.

### 2.3.2 Triplet Pairs Selection

Given a mini-batch of input units, SITE selects six units according to the propensity scores. Propensity score is the probability that a unit receives the treatment [28, 22]. For unit $i$, the propensity score $s_i$ is defined as $s_i = P(t_i = 1 | \mathbf{X} = \mathbf{x}_i)$. It's obvious that $s_i \in [0, 1]$. If $s_i$ is close to 1, more treated units should be distributed around the unit $i$ in the covariate space. Analogously, if $s_i$ is close to 0, more control units are available near the unit $i$. Moreover, if $s_i$ is close to 0.5, a mixture of both control and treated units can be found around the unit $i$. Thus, propensity score can kind of reflect the relative location of units in the covariate space, and we choose it as the indicator to select six data points. We use the logistic regression to calculate the propensity score [27].

Selecting three pairs of units in each mini-batch involves three steps, as shown in the left part of Figure 2.

• Step 1: Choose data pair $(\mathbf{x}_{\hat{i}}, \mathbf{x}_{\hat{j}})$ s.t.

$$(\hat{i}, \hat{j}) = \underset{i \in \mathcal{T}, j \in \mathcal{C}}{\operatorname{argmin}} |s_i - 0.5| + |s_j - 0.5|, \tag{3}$$

where $\mathcal{T}$ and $\mathcal{C}$ denote the treated group and control group, respectively. $\mathbf{x}_{\hat{i}}$ and $\mathbf{x}_{\hat{j}}$ are the closest units in the intermediate region where both control and treated units are mixed.

• Step 2: Choose $(\mathbf{x}_{\hat{k}}, \mathbf{x}_{\hat{l}})$ s.t.

$$\hat{k} = \underset{k \in \mathcal{C}}{\operatorname{argmax}} |s_k - s_{\hat{i}}|, \quad \hat{l} = \underset{l}{\operatorname{argmax}} |s_l - s_{\hat{k}}|. \tag{4}$$

$\mathbf{x}_{\hat{k}}$ is the farthest control unit from $\mathbf{x}_{\hat{i}}$, and is on the margin of control group with plenty of control units.

● Step 3: Choose $(\mathbf{x}_{\hat{m}}, \mathbf{x}_{\hat{n}})$ s.t.

$$\hat{m} = \underset{m \in \mathcal{T}}{\operatorname{argmax}} |s_m - s_{\hat{j}}|, \quad \hat{n} = \underset{n}{\operatorname{argmax}} |s_n - s_{\hat{m}}|. \tag{5}$$

$\mathbf{x}_{\hat{k}}$ is the farthest control unit from $\mathbf{x}_{\hat{i}}$, and is on the margin of control group with plenty of control units.

The pair $(\hat{i}, \hat{j})$ lies in the intermediate region of control and treated groups. Pairs $(\hat{k}, \hat{l})$ and $(\hat{m}, \hat{n})$ are located on the margins that are far away from the intermediate region. The selected triplet pairs can be viewed as hard cases. Intuitively, if the desired property of *preserved similarity* can be achieved for the hard cases, it will hold for other cases as well. Thus, we focus on preserving such a property for the hard cases (e.g., triplet pairs) in the latent space, and employ PDDM to achieve this goal.

### 2.3.3 Position-Dependent Deep Metric (PDDM)

PDDM was originally proposed to address the hard sample mining problem in image classification [14]. We adapt this design to the counterfactual estimation problem. In SITE, the PDDM component measures the local similarity of two units based on their relative and absolute positions in the latent space $\mathcal{Z}$.

The PDDM learns a metric that makes the local similarity of $(\mathbf{z}_i, \mathbf{z}_j)$ in the latent space close to their similarity in the original space. The similarity $\hat{S}(i, j)$ is defined as:

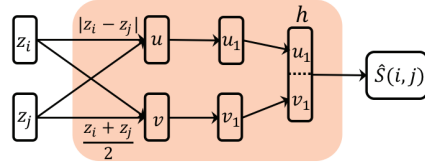

Figure 3: PDDM Structure.

$$\hat{S}(i, j) = \mathbf{W}_s \mathbf{h} + b_s, \tag{6}$$

where $\mathbf{h} = \sigma(\mathbf{W}_c[\frac{\mathbf{u}_1}{||\mathbf{u}_1||_2}, \frac{\mathbf{v}_1}{||\mathbf{v}_1||_2}]^T + b_c)$, $\mathbf{u} = |\mathbf{z}_i - \mathbf{z}_j|$, $\mathbf{v} = \frac{|\mathbf{z}_i + \mathbf{z}_j|}{2}$, $\mathbf{u}_1 = \sigma(\mathbf{W}_u \frac{\mathbf{u}}{||\mathbf{u}||_2} + b_u)$, $\mathbf{v}_1 = \sigma(\mathbf{W}_v \frac{\mathbf{v}}{||\mathbf{v}||_2} + b_v)$. $\mathbf{W}_c$, $\mathbf{W}_s$, $\mathbf{W}_v$, $\mathbf{W}_u$, $b_c$, $b_s$, $b_v$ and $b_u$ are the model parameters. $\sigma(\cdot)$ is a nonlinear function such as ReLU. As shown in Figure 3, the PDDM structure first calculates the feature mean vector $\mathbf{v}$ and the absolute position vector $u$ of the input $(\mathbf{z}_i, \mathbf{z}_j)$, and then feeds $\mathbf{v}$ and $\mathbf{u}$ to the fully connected layers separately. After normalization, PDDM concatenates the learned vectors $\mathbf{u}_1$ and $\mathbf{v}_1$, and feeds it to another fully connected layer to get the vector $\mathbf{h}$. The final similarity score $\hat{S}(,)$ is calculated by mapping the score $h$ to the $\mathcal{R}^1$ space.

The loss function of PDDM is as follows:

$$\mathcal{L}_{\text{PDDM}} = \frac{1}{5} \sum_{\hat{i},\hat{j},\hat{k},\hat{l},\hat{m},\hat{n}} \big[ (\hat{S}(\hat{k}, \hat{l}) - S(\hat{k}, \hat{l}))^2 + (\hat{S}(\hat{m}, \hat{n}) - S(\hat{m}, \hat{n}))^2 + (\hat{S}(\hat{k}, \hat{m}) - S(\hat{k}, \hat{m}))^2$$
$$+ (\hat{S}(\hat{i}, \hat{m}) - S(\hat{i}, \hat{m}))^2 + (\hat{S}(\hat{j}, \hat{k}) - S(\hat{j}, \hat{l}))^2 \big], \tag{7}$$

where $S(i, j) = 0.75|\frac{s_i + s_j}{2} - 0.5| - |\frac{s_i - s_j}{2}| + 0.5$. Similar to the design of the PDDM structure, the true similarity score $S(i, j)$ is calculated using the mean and the difference of two propensity scores. The loss function $\mathcal{L}_{\text{PDDM}}$ measures the similarity loss on five pairs in each mini batch: the pairs located in the margin area of the mini batch, i.e., $(\mathbf{z}_k, \mathbf{z}_l)$ and $(\mathbf{z}_m, \mathbf{z}_n)$; the pair that is most dissimilar among the selected points, i.e., $(\mathbf{z}_k, \mathbf{z}_m)$; the pairs located in the margin of the control/treated group, i.e., $(\mathbf{z}_j, \mathbf{z}_k)$ and $(\mathbf{z}_i, \mathbf{z}_m)$. As shown in Figure 2, minimizing $\mathcal{L}_{\text{PDDM}}$ on the above five pairs helps to preserve the similarity when mapping the original data into the representation space.

By using the PDDM structure, the similarity information within and between each of the pairs $(\mathbf{z}_{\hat{k}}, \mathbf{z}_{\hat{l}})$, $(\mathbf{z}_{\hat{m}}, \mathbf{z}_{\hat{n}})$, and $(\mathbf{z}_{\hat{k}}, \mathbf{z}_{\hat{n}})$ will be preserved.

### 2.3.4 Middle Point Distance Minimization (MPDM)

To achieve balanced distributions in the latent space, we design the middle point distance minimization (MPDM) component in SITE. MPDM makes the middle point of $(\mathbf{z}_{\hat{i}}, \mathbf{z}_{\hat{m}})$ close to the middle point of $(\mathbf{z}_{\hat{j}}, \mathbf{z}_{\hat{k}})$. The units $\mathbf{z}_{\hat{i}}$ and $\mathbf{z}_{\hat{j}}$ are located in a region where the control and treated units are sufficient and mixed. In other words, they are the closest units from treated and control groups

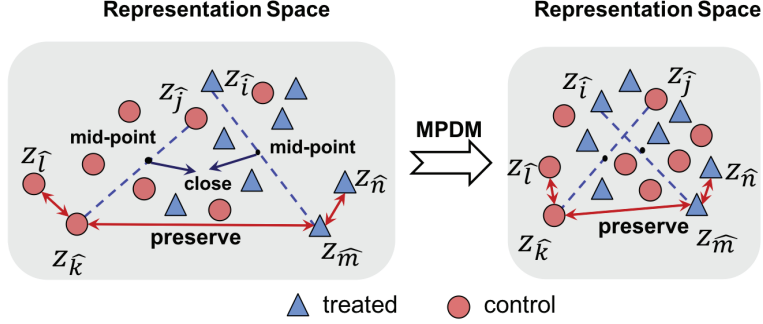

Figure 4: The effect of balancing distributions and preserving local similarity by using the proposed SITE method.

separately that lie in the intermediate zone. Meanwhile, $\mathbf{z}_{\hat{k}}$ is the farthest control unit from the margin of the treated group, and $\mathbf{z}_{\hat{m}}$ is the farthest treated unit from the margin of control group. We use the middle points of $(\mathbf{z}_{\hat{i}}, \mathbf{z}_{\hat{m}})$ and $(\mathbf{z}_{\hat{j}}, \mathbf{z}_{\hat{k}})$ to approximate the centers of treated and control groups, respectively. By minimizing the distance of two middle points, the units in the margin area are gradually made close to the intermediate region. As a result, the distributions of two groups will be balanced.

The loss function of MPDM is as follows:

$$\mathcal{L}_{\text{MPDM}} = \sum_{\hat{i},\hat{j},\hat{k},\hat{m}} \left( \frac{\mathbf{z}_{\hat{i}}+\mathbf{z}_{\hat{m}}}{2} - \frac{\mathbf{z}_{\hat{j}}+\mathbf{z}_{\hat{k}}}{2} \right)^2. \tag{8}$$

The MPDM balances the distributions of two groups in the latent space, while the PDDM preserves the local similarity. A 2-D toy example shown in Figure 4 vividly demonstrates the combined effect of MPDM and PDDM. Four units $\mathbf{x}_{\hat{i}}$, $\mathbf{x}_{\hat{j}}$, $\mathbf{x}_{\hat{k}}$ and $\mathbf{x}_{\hat{m}}$ are the same as what we choose in Figure 2. Figure 4 shows that MPDM makes the units that belong to treated group close to the control group, and PDDM restricts the way that the two groups close to each other. PDDM preserves the similarity information between $\mathbf{x}_{\hat{k}}$ and $\mathbf{x}_{\hat{m}}$. $\mathbf{x}_{\hat{k}}$ and $\mathbf{x}_{\hat{m}}$ are the farthest data points in the treated and control groups. When MPDM makes two groups approaching each other, PDDM ensures that the data points $\mathbf{x}_{\hat{k}}$ and $\mathbf{x}_{\hat{m}}$ are still the farthest, which prevents MPDM squeezing all data points into one point.

### 2.3.5 Outcome Prediction Network

With the components PDDM and MPDM, SITE is able to learn latent representations $z_i$ that balance the distributions of treated/control groups and preserve the local similarity of units in the original covariate space. Finally, the outcome prediction network is employed to estimate the outcome $\hat{y}_{t_i}^{(i)}$ by taking $z_i$ as input. Let $g(\cdot)$ denote the function learned by the outcome prediction network. We have $\hat{y}_{t_i}^{(i)} = g(z_i, t_i) = g(f(x_i), t_i)$.

The factual loss function is as follows:

$$\mathcal{L}_{\text{FL}} = \sum_{i=1}^{N} (\hat{y}_{t_i}^{(i)} - y_{t_i}^{(i)})^2 = \sum_{i=1}^{N} (g(f(x_i), t_i) - y_{t_i}^{(i)})^2, \tag{9}$$

where $y_{t_i}^{(i)}$ is the observed outcome.

### 2.3.6 Implementation and Joint Optimization

The representation network and outcome prediction network are standard feed-forward neural networks with Dropout [32] and ReLU activation function. The overall loss function of SITE in Eq.(2) can be jointly optimized. Adam [19] is adopted to solve the optimization problem. The PDDM and MPDM are calculated on triplet pairs during every batch.

Table 1: Performance comparison on IHDP and Jobs Dataset.

| Method | IHDP ($\sqrt{\mathcal{E}_{\text{PEHE}}}$) | | Jobs ($\mathcal{R}_{\text{pol}}$) | |
| --- | --- | --- | --- | --- |
| | Within-sample | Out-of-sample | Within-sample | Out-of-sample |
| OLS/LR$_1$ | $10.761 \pm 4.350$ | $7.345 \pm 2.914$ | $0.310 \pm 0.017$ | $0.279 \pm 0.067$ |
| OLS/LR$_2$ | $10.280 \pm 3.794$ | $5.245 \pm 0.986$ | $0.228 \pm 0.012$ | $0.733 \pm 0.103$ |
| HSIC-NNM [5] | $2.439 \pm 0.445$ | $2.401 \pm 0.367$ | $0.291 \pm 0.019$ | $0.311 \pm 0.069$ |
| PSM [27] | $7.188 \pm 2.679$ | $7.290 \pm 3.389$ | $0.292 \pm 0.019$ | $0.307 \pm 0.053$ |
| k-NN [8] | $4.432 \pm 2.345$ | $4.303 \pm 2.077$ | $0.230 \pm 0.016$ | $0.262 \pm 0.038$ |
| Causal Forest [33] | $4.732 \pm 2.974$ | $4.095 \pm 2.528$ | $0.232 \pm 0.018$ | $0.224 \pm 0.034$ |
| BNN [18] | $3.827 \pm 2.044$ | $4.874 \pm 2.850$ | $0.232 \pm 0.008$ | $0.240 \pm 0.012$ |
| TARNet [30] | $0.729 \pm 0.088$ | $1.342 \pm 0.597$ | $0.228 \pm 0.004$ | $0.234 \pm 0.012$ |
| CFR-MMD [30] | $0.663 \pm 0.068$ | $1.202 \pm 0.550$ | $\mathbf{0.213 \pm 0.006}$ | $0.231 \pm 0.009$ |
| CFR-WASS [30] | $0.649 \pm 0.089$ | $1.152 \pm 0.527$ | $0.225 \pm 0.004$ | $0.225 \pm 0.010$ |
| SITE (Ours) | $\mathbf{0.604 \pm 0.093}$ | $\mathbf{0.656 \pm 0.108}$ | $0.224 \pm 0.004$ | $\mathbf{0.219 \pm 0.009}$ |

## 3 Experiment

### 3.1 Experiment on Real Dataset

**Datasets.** Due to the missing counterfactual outcomes in reality, it is hard to measure the individual treatment effect estimation on traditional observational datasets. In order to evaluate the proposed method, we conduct the experiment on three datasets with different settings. IHDP and Jobs dataset are adopted in [30], one of the state-of-art methods. IHDP dataset aims to estimate the effect of specialist home visits on infant's future cognitive test scores, and Jobs dataset aims to estimate the effect of job training on employee status. Details about the IHDP and Jobs datasets are provided in the supplementary material. The twins dataset comes from the all twins birth in the USA between $1989 - 1991$ [2]. We focus on the same sex twin-pairs whose weights are less than $2000g$. Each record contains $40$ pre-treatment covariates related to the parents, the pregnancy and the birth. The treatment $T = 1$ is viewed as being the heavier one in the twins, and $T = 0$ is being the lighter one. The outcome is the mortality after one year. After eliminating the records containing missing features, the final dataset contains $5409$ records. In this setting, both treated and control outcomes can be observed. In order to create the selection bias, we execute the following procedures to selectively choose one of the twins as the observation and hide the other: $T_i|\mathbf{x}_i \sim \text{Bern}(\text{Sigmoid}(\mathbf{w}^T\mathbf{x} + n))$, where $\mathbf{w}^T \sim \mathcal{U}((-0.1, 0.1)^{40 \times 1})$ and $n \sim \mathcal{N}(0, 0.1)$.

**Baselines.** We compare the proposed method with the following three groups of baselines. (1) Regression based methods: Least square Regression with the treatment as feature (**OLS/LR$_1$**), separate linear regressors for each treatment group (**OLS/LR$_2$**); (2) Nearest neighbor matching based methods: Hilbert-Schmidt Independence Criterion based Nearest Neighbor Matching (**HSIC-NNM**) [5], Propensity score match with logistic regression (**PSM**) [27], k-nearest neighbor (**k-NN**) [8]; (3) Tree and forest based method: Causal Forest [33]. (4) Representation learning based methods: Balancing neural network (**BNN**) [18], counterfactual regression with MMD metric (**CFR-MMD**) [30], counterfactual regression with Wasserstein metric (**CFR-WASS**) [30], and Treatment-Agnostic Representation Network (**TARNet**) [30].

**Performance Measurement.** On IHDP dataset, the Precision in Estimation of Heterogeneous Effect ($\mathcal{E}_{\text{PEHE}}$) [13] is adopted as the performance metric, where $\mathcal{E}_{\text{PEHE}} = \frac{1}{N}\sum_{i=1}^{N}\left(\mathbb{E}_{(y_0^{(i)}, y_1^{(i)}) \sim \mathcal{P}_{\mathbf{Y}|\mathbf{x}_i}}\left[y_0^{(i)} - y_1^{(i)}\right] - (\hat{y}_0^{(i)} - \hat{y}_1^{(i)}))^2$; On jobs dataset, the policy risk $\mathcal{R}_{\text{pol}}$ [30] is used as the metric, which is defined as: $\mathcal{R}_{pol} = 1 - \left(\mathbb{E}[\mathbf{Y}_1|\pi(x) = 1]\mathcal{P}(\pi(x) = 1) + \mathbb{E}[\mathbf{Y}_0|\pi(x) = 0]\mathcal{P}(\pi(x) = 0)\right)$, where $\pi(x) = 1$ if $\hat{y}_1 - \hat{y}_0 > 0$ and $\pi(x) = 0$, otherwise. The policy risk measures the expected loss if the treatment is taken according to the ITE estimation. For PEHE and policy risk, the smaller value is, the better the performance. On the Twins dataset, the class is imbalanced, so we adopt area over ROC curve(AUC) on outcomes as the performance measure, as suggested in [25]. The larger AUC is, the better the performance.

On each dataset, we consider both the within-sample case and out-of-sample case [30]. In the former case, the observed outcome is available, while in the latter case, only the pre-treatment covariates are available. In the within-sample case, the performance metric is measured on the training dataset, and the out-of-sample case is on the test dataset. Since we never use the ground truth ITE during the training procedure, performance metric is a meaningful metric in both the within-sample and out-of-sample cases.

**Results Analysis** [2]. Tables 1 and 2 show the performance of 10 realizations of our method and baselines on three datasets. SITE achieves the best performance on the IHDP and Twins datasets, and on the Jobs dataset, SITE achieves similar results to the best baseline. It confirms that preserving the local similarity information during representation learning can help better estimate the counterfactual outcomes and ITE.

Generally speaking, the representation learning based methods perform better than the linear regression based and nearest neighbor matching based methods. The regression-based methods are not specially designed to deal with counterfactual inference, so the performance is affected by the selection bias. The nearest neighbor based methods incorporate the similarity information to overcome the selection bias, but they only use the observed outcomes of neighbors in the other group as their counterfactual outcomes, which might be inaccurate and unreliable.

Among the representation learning based methods, our proposed method outperforms all other baselines. The methods considering balancing distributions (BNN, CFR MMD, CFR WASS, and the proposed method) obtain better performance than the method without balancing property (TARNet). BNN balances the distributions of two treatment groups in the representation space and views the treatment $t_i$ as a feature. While TARNet doesn't have any regularization in the representation space, and its outcome prediction network is dichotomous. CFR-MMD

Table 2: Performance comparison on twins dataset.

|  | **Twins** (AUC) | |
| --- | --- | --- |
| Method | Within-sample | Out-of-sample |
| OLS/LR$_1$ | $0.660 \pm 0.005$ | $0.500 \pm 0.028$ |
| OLS/LR$_2$ | $0.660 \pm 0.004$ | $0.500 \pm 0.016$ |
| HSIC-NNM [5] | $0.762 \pm 0.011$ | $0.501 \pm 0.017$ |
| PSM [27] | $0.500 \pm 0.003$ | $0.506 \pm 0.011$ |
| k-NN [8] | $0.609 \pm 0.010$ | $0.492 \pm 0.012$ |
| BNN [18] | $0.690 \pm 0.008$ | $0.676 \pm 0.008$ |
| TARNet [30] | $0.849 \pm 0.002$ | $0.840 \pm 0.006$ |
| CFR-MMD [30] | $0.852 \pm 0.001$ | $0.840 \pm 0.006$ |
| CFR-WASS [30] | $0.850 \pm 0.002$ | $0.842 \pm 0.005$ |
| SITE (Ours) | $\mathbf{0.862 \pm 0.002}$ | $\mathbf{0.853 \pm 0.006}$ |

and CFR-WASS have the same dichotomous outcome prediction networks, but they use different integral probability metrics to balance the distributions. The results of BNN, CFR MMD, CFR WASS, and the proposed method SITE indicate that balancing the distributions of different treatment groups indeed helps reduce the negative effect of selection bias.

Compared with CFR-MMD and CFR-WASS, our proposed method SITE not only considers the balancing property (MPDM), but also preserves the local similarity information in the original feature space (PDDM). It is observed that on the IHDP dataset, SITE significantly improves the results in both within-sample case and out-of-sample case. On Jobs and Twins datasets, the performance of SITE are comparable with the best baseline. The results on three datasets demonstrate the effectiveness of preserving local similarity information in the latent space. Moreover, with the specifically designed PDDM and MPDM structures, SITE can efficiently calculate the similarity information and balance the distributions of different treatment groups. The PDDM and MPDM structures only require the selected triplet pairs, which avoids handling the entire dataset. By jointly considering distribution balancing and similarity preserving, the proposed method can effectively and efficiently estimate the individual treatment effect.

**Experiments on PDDM and MPDM.** PDDM (for local similarity preserving) and MPDM (for balancing) aim to reduce the generalization error when inferring the potential outcomes. As SITE assumes that similar units shall have similar treatment outcomes, PDDM and MPDM are able to preserve the local similarity information and meanwhile achieve the balanced distributions in the latent space. In order to further comfirm the effect of PDDM and MPDM, we compare SITE with SITE-without-PDDM and SITE-without-MPDM on all the three datasets. Table 3 shows the re-

sults. It can be observed that SITE outperforms the baselines without PDDM or MPDM structures. Therefore, the two structures, PDDM and MPDM, are necessary to improve the ITE estimation.

Table 3: Experiment on PDDM & MPDM: Performance Comparison on Three Datasets.

| Dataset | | SITE | SITE-without-PDDM | SITE-without-MPDM |
|---|---|---|---|---|
| **IHDP** ($\mathcal{E}_{\text{PEHE}}$) | Within-sample | **$0.604 \pm 0.093$** | $0.635 \pm 0.127$ | $0.859 \pm 0.093$ |
| | Out-of-sample | **$0.656 \pm 0.108$** | $0.685 \pm 0.128$ | $1.416 \pm 0.476$ |
| **Jobs** ($\mathcal{R}_{\text{pol}}$) | Within-sample | **$0.224 \pm 0.004$** | $0.233 \pm 0.004$ | $0.222 \pm 0.003$ |
| | Out-of-sample | **$0.219 \pm 0.009$** | $0.234 \pm 0.012$ | $0.234 \pm 0.009$ |
| **Twins** (AUC) | Within-sample | **$0.862 \pm 0.002$** | $0.770 \pm 0.033$ | $0.796 \pm 0.040$ |
| | Out-of-sample | **$0.853 \pm 0.006$** | $0.776 \pm 0.033$ | $0.788 \pm 0.040$ |

## 3.2  Experiment on Synthetic Dataset

**Data Generation.**  To evaluate the robustness of SITE, we design experiments on a synthetic dataset. Following the settings in [36], the synthetic data are generated as follows: we generate 5000 control samples from $N(0^{10\times1}, 0.5 \times (\Sigma + \Sigma^T))$ and 2500 treated samples from $N(\mu_1, 0.5 \times (\Sigma + \Sigma^T))$, where $\Sigma \sim U((-1,1)^{10\times10})$. By varying the value of $\mu_1$, data with different levels of selection bias are generated. Kullback-Leibler divergence (KL divergence) is adopted to measure the selection bias. The larger the KL divergence is, the smaller the overlapping of simulated control and treated groups is, and the larger the selection bias is. The outcome is generated as $\mathbf{y}|x \sim (\mathbf{w}^T x + n)$, where $\mathbf{w} \sim U((-1,1)^{10\times2})$, and $n \sim N(0^{2\times1}, 0.1 \times I^{2\times2})$.

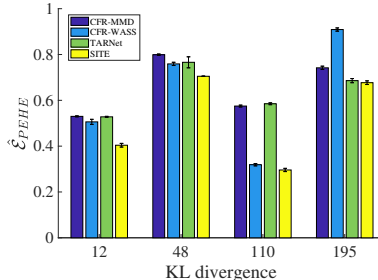

Figure 5: Performance Comparison on Synthetic Dataset.

**Result Analysis.** We compare the proposed method with the most competitive baselines, TARNet, CFR-MMD and CFR-WASS. The mean and variance of the $\mathcal{E}_{PEHE}$ on 10 realizations are reported in Figure 5. It is observed from the figure that SITE consistently outperforms baseline methods under different levels of divergence.

## 4  Conclusion

In this paper, we present an efficient deep representation learning method for estimating individual treatment effect. The proposed method jointly preserves the local similarity information and balances the distributions of control and treated groups. Experimental results on the IHDP, Jobs and Twins datasets show that, in most cases, our method achieves better performance than the state-of-the-art. Extensive evaluation of our method further validates the benefits of preserving local similarity in ITE estimation.

## 5  Acknowledgment

This work was supported in part by the US National Science Foundation under grants NSF IIS-1747614, IIS-1218393 and IIS-1514204. Any opinions, findings, and conclusions or recommendations expressed in this material are those of the author(s) and do not necessarily reflect the views of the National Science Foundation. Also, we gratefully acknowledge the support of NVIDIA Corporation with the donation of the Titan Xp GPU used for this research.

## Footnotes

[1]Some works [30] define ITE as the form of CATE: $\mathbf{ITE}_i = E(\mathbf{Y}_1^{(i)}|\mathbf{x}) - E(\mathbf{Y}_0^{(i)}|\mathbf{x})$.

[2]The code of SITE is available at `https://github.com/Osier-Yi/SITE`.

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
