[Supplementary Material]

# Supplementary Material for Representation Learning for Treatment Effect Estimation from Observational Data

## 1 Dataset

The following is the detailed description about the IHDP and Jobs datasets.

**IHDP.** The dataset is provided by Hill [3] based on the randomized controlled experiment conducted by Infant Health and Development Program. The program aims to estimate the effect of specialist home visits on infant's future cognitive test scores. A biased subset of the treated group is removed in order to create the selection bias. There are total $747$ records with $139$ treated records and $608$ control records in the dataset. Each record contains 25 pre-treatment covariates related to the children and their mothers. The outcomes are simulated by the setting "A" of the NPCI package [1].

**Jobs.** This dataset comprises Lalonde randomized controlled experiment (297 treated records and 425 control records) and the PSID observational group (2490 control records) [4, 6]. Each record contains 8 covariates, such as age, education, ethnicity, as well as previous earnings. The outcome is the employment status with/without job training.

## 2 Performance Metric and Experiment Settings

On the IHDP dataset, the distributions of potential outcomes are known, so we use the Precision in Estimation of Heterogeneous Effect (PEHE) [3] as the performance metric. The definition of PEHE is: $\mathcal{E}_{\text{PEHE}} = \frac{1}{N}\sum_{i=1}^{N}(\mathbb{E}_{(y_0^{(i)},y_1^{(i)})\sim\mathcal{P}_{\mathbf{Y}|\mathbf{x}_i}}\left[y_0^{(i)} - y_1^{(i)}\right] - (\hat{y}_0^{(i)} - \hat{y}_1^{(i)}))^2$, where $\hat{y}_0^{(i)}$ and $\hat{y}_1^{(i)}$ are the estimated control and treated outcomes of unit $i$, respectively. The lower the $\mathcal{E}_{\text{PEHE}}$ is, the better the method is. On IHDP dataset, we conduct experiment over 10 realizations of the outcomes with $63/27/10$ ratio of train/validation/test splits, as suggested in [5].

On the Jobs dataset, there is no ground truth ITE available. To evaluate the proposed method, the policy risk $\mathcal{R}_{\text{pol}}$ [5] is used as the metric, which is defined as: $\mathcal{R}_{pol} = 1 - \big(\mathbb{E}[\mathbf{Y}_1|\pi(x) = 1]\mathcal{P}(\pi(x) = 1) + \mathbb{E}[\mathbf{Y}_0|\pi(x) = 0]\mathcal{P}(\pi(x) = 0)\big)$, where $\pi(x) = 1$ if $\hat{y}_1 - \hat{y}_0 > 0$ and $\pi(x) = 0$, otherwise. The policy risk measure the expected loss if the treatment is taken according to the ITE estimation. The policy risk of Jobs dataset can be estimated from the randomized experiment part:

$$\hat{\mathcal{R}}_{\text{pol}} = 1 - \big(\frac{1}{|A_1 \cap T_1 \cap E|}\sum_{\mathbf{x}_i \in A_1 \cap T_1 \cap E} y_1^{(i)}\frac{|A_1 \cap E|}{|E|} + \frac{1}{|A_0 \cap T_0 \cap E|}\sum_{\mathbf{x}_i \in A_0 \cap T_0 \cap E} y_0^{(i)}\frac{|A_0 \cap E|}{|E|}\big),$$

(1)

where $E = \{\mathbf{x}_i : \mathbf{x}_i \text{ is from the randomized experiment.}\}$; $A_1 = \left\{\mathbf{x}_i : \hat{y}_1^{(i)} - \hat{y}_0^{(i)} > 0\right\}$; $A_0 = \left\{\mathbf{x}_i : \hat{y}_1^{(i)} - \hat{y}_0^{(i)} < 0\right\}$; $T_1 = \{\mathbf{x}_i : t_i = 1\}$; $T_0 = \{\mathbf{x}_i : t_i = 0\}$. The lower value of $\hat{\mathcal{R}}_{\text{pol}}$ indicates the ITE estimation method can better support the decision strategy. On the Jobs dataset, we average over 10 train/validation/test splits with $56/24/20$ split ratio, as suggested in [5].

Table 1: Hyper-parameter searching space

| Hyper-parameter | Range |
|---|---|
| $\beta, \gamma$ | $\left\{10^{k/2}\right\}_{k=-6}^{6}$ |
| $\lambda$ | $10^{-4}$ |
| Number of hidden layers in the representation network | 1, 2, 3 |
| Number of hidden layers in outcome prediction network | 1, 2, 3 |
| dimension of each layer in the representation network | 50,100,200 |
| dimension of each layer in the outcome prediction network | 50,100,200 |
| batch size | 50,100 |

Table 2: Performance on three datasets.

| Method | IHDP ($\mathcal{E}_{\text{PEHE}}$) | | Jobs ($\mathcal{R}_{\text{pol}}$) | | Twins (AUC) | |
|---|---|---|---|---|---|---|
| | Within-sample | Out-of-sample | Within-sample | Out-of-sample | Within-sample | Out-of-sample |
| SITE-MMD | $1.162 \pm .118$ | $1.242 \pm .163$ | $.194 \pm .015$ | $.218 \pm .010$ | $.710 \pm .003$ | $.705 \pm .006$ |
| SITE-WASS | $.993 \pm .112$ | $1.459 \pm .481$ | $.190 \pm .015$ | $.232 \pm .011$ | $.849 \pm .003$ | $.762 \pm .007$ |

## 3 Hyper-parameter Optimization

Table summarizes the searching space of the hyper-parameters. In the PDDM structure, $\mathbf{W}_u$ and $\mathbf{W}_v \in \mathcal{R}^{d \times d}$; $\mathbf{W}_u \in \mathcal{R}^{2d \times d}$ and $\mathbf{W}_s \in \mathcal{R}^{1 \times d}$, where $d$ is the number of the representation layer. The parameters of baselines are set as what is suggested in the original papers.

Due to the fact of missing counterfactuals, we cannot directly use the $\mathcal{E}_{PEHE}$, and AUC on the validation dataset as the selection criterion. Fortunately, the underlying assumption of SITE, that there exits a model that can work well on both control and treated group, makes the RMSE of the observed outcome on the validation dataset to be a reasonable hyper-parameter selection criterion.

## 4 Methodology Discussion

There are a lot of alternatives to do balancing. Here we replace the balancing structure MPDM with the distribution metric such as Wasserstein metric [2], and Maximum Mean Discrepancy(MMD) [7]. The loss for balancing is as follows: $L_B = \mathcal{D}(\mathcal{Z}_T, \mathcal{Z}_C)$, where $\mathcal{D}(\cdot, \cdot)$ denotes the distribution metric. And the final loss function is as follows: The loss function is as follows:

$$\mathcal{L} = \mathcal{L}_{\text{FL}} + \beta \mathcal{L}_{\text{PDDM}} + \alpha \mathcal{L}_{\text{B}} + \lambda ||W||_2. \tag{2}$$

SITE with MMD as the distribution metric is denoted as SITE-MMD and with Wasserstein metric is denoted as SITE-WASS. The performance of SITE-MMD and SITE-WASS is reported in the table 2.

## 5 Representation Visualization

The t-SNE visualization of the learned representation on the synthetic dataset is shown in Figure 1. The synthetic dataset is the same as what we generated in Section 3.2. The visualization of SITE confirms that the MPDM minimizes the distance between the approximated centers of two groups.