[Reviews · NeurIPS 2018]

Reviewer 1



This paper takes on the difficult task of estimating individual treatment effects in observational data. The authors propose a framework, SITE, which both ensures global balance and local similarity using deep representation learning. Results are demonstrated on benchmark datasets. This is a high quality paper. I have some quibbles and concerns with it, but it is tackling a difficult problem in an interesting way. I realize that the authors are building on an existing literature in computer science on the subject (e.g. Shalit et al) but I don’t think framing this as estimation of the individual treatment effect is quite right. The individual treatment effect is completely intractable with the standard assumptions that are typically used, because there is only the one observation. What the paper really estimates is heterogeneous treatment effects for high-dimensional covariates x. A symptom of this issue comes in equation 1. The authors invoke the potential outcomes framework and define the ITE in terms of E[Y_1(i) | x_i] etc. but what’s the expectation over? In the potential outcomes framework, the potential outcomes aren’t random and its not clear what the conditioning is doing. At some level this is just a squabble over notation but I think it points to some subtleties in what is meant by individual treatment effect. I also have some concerns about the stated assumptions. It seems in addition to assumptions 2.1 and 2.2, there is some kind of SUTVA or consistency assumption needed, right? I also think that the authors are not taking seriously enough the strength of the positivity assumption. Presumably enforcing local similarity would not be necessary if positivity straightforwardly held. In other words, if you weren’t trying to smooth over a high-dimensional space, you wouldn’t need the latent representation in the first place. This general point has been made in the high dimensional case by a recent paper by D’Amour et al (https://arxiv.org/abs/1711.02582) but I think it also applies here in these relatively low-dimensional settings because the estimand is so specific to a narrow region of the space. Ultimately these quibbles are fairly small though. The core idea here seems to be a good one and the paper cleverly brings representation learning into causal inference which is both original and likely to have a significant impact by bringing two distinct communities closer together. I found the paper to be quite clear although I do worry that there is a relatively small group of scholars who have training in both causal inference and deep learning such that they can properly appreciate this work. That may ultimately be one of the challenges of bridging communities like this though. *** Response to Author Feeedback*** I thank the authors for the feedback and the additional study. In going through the other reviews, it is clear to me that Reviewer 3 and I essentially agree on the weaknesses of the paper and the way the response memo does and doesn't address them. The difference in scores is attributable primarily to how strongly we feel about those weaknesses. The primary concern with the paper is the clarity both in the analysis of the experimental results (although I agree that the PDDM study in the memo is great and should be added) and in the estimands/assumptions. I followed R3's excellent example and also revisited the prior work. I still disagree about framing this as an individual treatment effect rather than a conditional average treatment effect - and notice further that the ITE is actually defined as a CATE in one of the prior papers (which I still disagree with, but that correspondence might be worth referencing). Obviously space is always at a premium but I think the work will have greater impact if you can summarize some of the key elements of the prior work you are building on. I look forward to the discussion of the recent work on positivity. I also do encourage you to explicitly add the consistency/SUTVA assumption for completeness.

Reviewer 2



This paper addresses the estimation of causal relationships or causal effects of interventions on outcomes but at an individual level from observational data. This problem is quite hard as it requires the estimation of counterfactuals that are by definition never observed at the individual level. It has received some attention recently (e.g., [17], [28]) and this paper does extend the state of the art in my opinion. The main contribution is to leverage and preserve local similarity information in the deep learning representation process to estimate individual treatment effect. The approach is also able to balance the distributions of the control and treatment groups effectively. The approach proposed by the authors is sound although the presentation of the methodology can be improved in my opinion. There are concepts introduced early on in the paper that only make sense a few page after (e.g., PDDM and MPDM). The paper could benefit from a more intuitive presentation of these concepts early on so that it does not let the reader hanging. Also, in section 2.3.3 line 170, I would recommend explaining further the structure of the loss function \cal{L}_{PDDM}. The results presented by the authors are convincing. They have compared their approach against the state of the art and experiments are showing that that there SITE techniques does better than the current state of the art to estimate individual treatment effects.

Reviewer 3



# Summary This paper proposes a method for estimating individual treatment effects. The basic approach is to propose a regression y_i = g(f(x_i), t_i) where f(x_i) is a learned representation of the covariates, and g predicts the response using this representation and the treatment. The main novelty is in the latent representation g(x_i). The given motivation is that points that are close in the covariate space should also be close in the representation space. This is achieved by adding two terms to the mini-batch loss function. # Strengths The problem is self evidently important, the idea is creative, and seems to work well under to an (apparently) broad battery of empirical tests. # Weaknesses I had trouble understanding this paper, but I'm not sure if that's due to the exposition or because it's a little ways outside my normal area of expertise. In particular, I don't immediately see the answer to: 1. When, if ever, is SITE a consistent estimator of the true average ITE? What assumptions are required for the estimator to be a valid estimator of the causal effect? In particular, the loss function is only defined over minibatches, so the objective function for the regression is left implicit. Presupposing that SGD does in fact converge, does this oddball objective function cause any issues? 2. Is the intuitive motivation (representation preserves local similarity) actually supported by the empirical results? It's clear that SITE does somewhat better than the comparitor methods, but it's not clear to me why this is---could it attributed simply to higher model capacity? Experiments showing that SITE actually learns representations that do a better job of preserving local similarity, and that this matters, would strengthen the paper. I note that the authors claim (line 263-265) that the experiments show this already, but I don't understand how. More broadly, the whole procedure in this paper feels a bit ad hoc. It's not clear to me which parts of the model are really important for the results to hold. # Overview Based on the above critiques, I've voted (weakly) to reject. However, if good answers to the above points of confusion can be incorporated into the text, I'd be happy to reconsider. # Further questions and comments 1. The learning procedure described in the paper seems to take propensity scores as inputs. E.g. triplet pair selection relies on this. How are the required propensity scores determined? 2. Does the SGD actually converge? It seems possible that selecting triplets within the mini-batch may lead to issues. 3. What is the summation (\Sum) in equation 7 over? I thought there was exactly one \hat{i}, \hat{j}, etc. per minibatch 4. Is the correct interpretation of the experiment in section 3.2 that the covariates x are generated according to one of the two normal distributions, the response is generated independent of T, and the ground truth ITE is 0? Assuming this is correct, the text should be rewritten to be clearer. I'm also unclear what the inputs to the KL distance are, and why KL is chosen as the distance #### Update I thank the authors for responding to my comments and questions. 1. The ablation study with PDDM is helpful. I think the paper would be improved by adding this, and also one for MPDM loss term. The presented results show that PDDM improves the ITE estimate, but, as far as I can see, it does not show that the PDDM term actually causes the learned representations to better preserve local similarity. 2. I'm still confused by when, if ever, the method described in the paper is a correct estimator for ITE (defining 'correct' sensibly is part of the challenge here). The author response didn't cleared this up for me. For example, it was still unclear why assumptions 2.1 and 2.2 are included given that, as far as I can tell, neither is actually referenced again in the paper. I was sufficiently confused by this that I went back and read the related prior work ("Learning Representations for Counterfactual Inference" Johansson, Shalit, Sontag, and "Estimating Individual Treatment Effect: Generalization Bounds and Algorithms", ref [28] in the paper). These papers establish a clear meaning for the use of representation learning for causal inference: the PEHE is bounded by an expression that includes the distance between P(r | t=0) and P(r | t=1), the distributions over the representations for the treated and untreated. I'm convinced this could provide a clear motivation and (hand wavy) justification for SITE. However, I don't think the paper explains this well---if the method is justified by ref [28] then the key results and the connection should be summarized in the paper! There is presumably some analogous idea making use of the assumption that similar units will have similar responses, but I'm again unsure what such a result is or exactly how it might apply to the proposed method. Although I'm now convinced that the method can probably be justified properly, I don't think this is done in the paper, and I view the the confusing presentation as a serious problem. For this reason, I'm still voting to reject. However, I note that reviewer 1 has essentially the same criticism (to wit: this method doesn't seem to actually estimate ITE), but doesn't view this is a serious issue.